# Negative Control of Cell Migration by Rac1b in Highly Metastatic Pancreatic Cancer Cells Is Mediated by Sequential Induction of Nonactivated Smad3 and Biglycan

**DOI:** 10.3390/cancers11121959

**Published:** 2019-12-06

**Authors:** Hannah Otterbein, Hendrik Lehnert, Hendrik Ungefroren

**Affiliations:** 1First Department of Medicine, University Hospital Schleswig-Holstein, Campus Lübeck, D-23538 Lübeck, Germany; hannahotterbein@web.de (H.O.); hendrik.lehnert@uni-luebeck.de (H.L.); 2Clinic for General Surgery, Visceral, Thoracic, Transplantation and Pediatric Surgery, University Hospital Schleswig-Holstein, Campus Kiel, D-24105 Kiel, Germany

**Keywords:** biglycan, chemokinesis, migration, pancreatic ductal adenocarcinoma, RAC1B, SMAD3, transforming growth factor-beta

## Abstract

Expression of the small GTPase, Ras-related C3 botulinum toxin substrate 1B (RAC1B), a RAC1-related member of the Rho GTPase family, in tumor tissues of pancreatic ductal adenocarcinoma (PDAC) has been shown previously to correlate positively with patient survival, but the underlying mechanism(s) and the target genes involved have remained elusive. Screening of a panel of established PDAC-derived cell lines by immunoblotting indicated that both RAC1B and Mothers against decapentaplegic homolog 3 (SMAD3) were more abundantly expressed in poorly metastatic and well-differentiated lines as opposed to highly metastatic, poorly differentiated ones. Both siRNA-mediated RAC1B knockdown in the transforming growth factor (TGF)-β-sensitive PDAC-derived cell lines, Panc1 and PaCa3, or CRISPR/Cas-mediated knockout of exon 3b of *RAC1* in Panc1 cells resulted in a dramatic decrease in the expression of *SMAD3*. Unexpectedly, the knockdown of *SMAD3* reproduced the promigratory activity of a RAC1B knockdown in Panc1 and PaCa3, but not in TGF-β-resistant BxPC3 and Capan1 cells, while forced expression of SMAD3 alone was able to mimic the antimigratory effect of ectopic RAC1B overexpression in Panc1 cells. Moreover, overexpression of SMAD3 was able to rescue Panc1 cells from the RAC1B knockdown-induced increase in cell migration, while knockdown of SMAD3 prevented the RAC1B overexpression-induced decrease in cell migration. Using pharmacological and dominant-negative inhibition of SMAD3 C-terminal phosphorylation, we further show that the migration-inhibiting effect of SMAD3 is independent of its activation by TGF-β. Finally, we provide evidence that the antimigratory program of RAC1B-SMAD3 in Panc1 cells is executed through upregulation of the migration and TGF-β inhibitor, biglycan (BGN). Together, our data suggest that a RAC1B-SMAD3-BGN axis negatively controls cell migration and that SMAD3 can induce antimigratory genes, i.e., *BGN* independent of its role as a signal transducer for TGF-β. Therefore, targeting this novel pathway for activation is a potential therapeutic strategy in highly metastatic PDAC to interfere with invasion and metastasis.

## 1. Introduction

Pancreatic ductal adenocarcinoma (PDAC) is a highly aggressive tumor with an extremely poor prognosis. In Western countries, PDAC ranks fourth in the order of tumor-related deaths with an increasing prevalence [1]. The overall five-year survival rates of less than 5% result in part from the fact that tumors are usually detected at an advanced stage when the cancer cells have already metastasized [2]. The available therapeutic options, most of which are palliative, include surgery, radiation, chemotherapy, immunotherapy, and the use of biologicals. However, thus far, targeted therapies directed towards cancer-associated molecular pathways have not yielded satisfactory results [1].

The human *RAC1* gene encodes two known isoforms, termed RAC1 and RAC1B. RAC1B differs from RAC1 by an in-frame insertion of an additional exon (exon 3b) of 57 nucleotides. This results in profound alterations in its biochemical and signaling properties as well as in its cellular effects, some of which are antagonistic to those of RAC1 [3]. RAC1B has been reported to promote cellular proliferation and antiapoptosis, but unlike RAC1 its role in other cancer-related processes, such as epithelial–mesenchymal transition (EMT) and cell migration and invasion remains poorly characterized. RAC1 is known to promote EMT, migration and invasion [4,5] and the same role in EMT induced by matrix metalloprotease (MMP)3 has been postulated for RAC1B (reviewed in [3]). However, we observed opposite effects of endogenous RAC1B at least with regard to EMT and random cell migration (chemokinesis) triggered by transforming growth factor (TGF)-β1 in benign and malignant ductal epithelial cells of pancreatic [6,7] and breast [8] origin. The ability of RAC1B to block two tumor-promoting functions of TGF-β, EMT and cell motility in vitro is a significant observation in the light of recent data revealing that RAC1B protein expression was more abundant in a PDAC-derived cell line of low metastatic potential (Colo357) compared with a cell line of high metastatic potential (Panc1) and, even more striking, in PDAC tissues correlated with prolonged patient survival [6]. Our published data also suggest that RAC1B-mediated suppression of TGF-β1-dependent chemokinesis involves attenuation of non-Smad signaling, i.e., by p38 mitogen-activated protein kinase (MAPK) and ERK1/2 MAPK [7], the activation of which is crucial for TGF-β1-induced EMT and cell motility. Using experimental strategies of RAC1B knockdown (KD) or knockout (KO) in PDAC-derived Panc1 and breast cancer-derived MDA-MB-231 cells, we have recently shown that the inhibition of TGF-β1-dependent chemokinesis by RAC1B was a consequence of downregulation of the TGF-β type I receptor, activin receptor-like kinase (ALK)5 [9]. However, despite some evidence for a protective role of RAC1B in PDAC development it is not known whether this is due to its ability to block TGF-β-induced EMT or whether RAC1B operates through a TGF-β-independent mechanism.

In the current study, we focused our attention on the role of RAC1B in the regulation of basal migratory/chemokinetic activity in PDAC, employing pancreatic carcinoma cells of different metastatic potential. Since invasion and metastasis has been shown to correlate positively with autocrine TGF-β signaling [10,11], we included both cell lines that are resistant to TGF-β due to lack of protein expression of either SMAD4, i.e., BxPC3, Capan1, and Capan2 [11,12,13,14], or TGF-β type II receptor (MiaPaCa2) [13,15] as well as others that are *DPC4* wildtype and have retained sensitivity to this growth factor such as Colo357, PaCa3, and Panc1 [7,9,10,12,13,14]. Prompted by the finding of a decrease in the expression of SMAD3, a central intracellular signal transducer of TGF-β, following RAC1B knockdown or knockout, we studied the possible role of SMAD3 as a mediator of RAC1B’s antimigratory function. Following phosphorylation at the C-terminal Ser-Ser-X-Ser motif (pSMAD-C) by the ALK5 kinase, the receptor-regulated SMADs, SMAD2 and SMAD3, become activated and transduce the signal from ALK5 binding ligands (TGF-β, activin, myostatin) to the nucleus after complex formation with the common-mediator SMAD, SMAD4. Here, the pSMAD3-C-SMAD4 complex can bind directly to the promoters of TGF-β target genes to regulate gene expression and more complex cellular responses such as growth inhibition, EMT, cell migration and invasion. Hence, when acting as a signal transducer of TGF-β, SMAD3 clearly has a tumor-promoting role. However, whether SMAD3 also impacts cellular functions, in particular cell motility, in its non-C-terminally phosphorylated (non-pSmad3-C) state and, if so, whether these contrast with pSMAD3-C-mediated functions remains unknown. Here we provide strong experimental evidence for a C-terminal phosphorylation/activation-independent function of SMAD3, negative regulation of cell migration in response to induction by RAC1B.

## 2. Results

### 2.1. Expression Levels of RAC1B and SMAD3 are Inversely Correlated with the Metastatic Potential of PDAC-Derived Cell Lines

Given the known antimigratory role of RAC1B in Panc1 cells, we reasoned that RAC1B might be of more general importance in determining the migratory/invasive potential of PDAC-derived cells and hence might be enriched in poorly metastatic cells. Preliminary evidence for this assumption came from a recent study analysing the role of RAC1B in epithelial differentiation, where RAC1B protein was found to be more abundant in poorly metastatic Capan1 than in highly metastatic Panc1 cells [16]. To validate this on a broader scale, we screened by immunoblotting a battery of permanent PDAC lines that differ in their metastatic activities upon transplantation in severe combined immunodeficiency (SCID) mice [17]. Three cell lines with low metastatic potential, BxPC3 [17], Capan2 [17] and Colo357 [18,19], and another three cell lines with high metastatic potential, Panc1 [17], MiaPaCa2 ]17] and PaCa3 [20], were selected for this purpose. Intriguingly, RAC1B was abundantly expressed in well differentiated/poorly metastatic cell lines, whereas low RAC1B levels were observed in poorly differentiated/highly metastatic cell lines (Figure 1A). Moreover, in the same cell lines, we noted that SMAD3, a central intracellular signal transducer of TGF-β, mirrored the expression pattern of RAC1B but was not strictly correlated with sensitivity of the cells to TGF-β (Figure 1A). A very similar profile of RAC1B and SMAD3 expression among PDAC lines-i.e., high mRNA levels in poorly metastatic lines and low levels in highly metastatic lines-was observed in quantitative real-time RT-PCR (qPCR) analyses (Appendix A), suggesting that the differences in protein abundance result from corresponding changes in RAC1B and SMAD3 mRNA.

Permanent PDAC-derived cell lines differ in the spectrum of mutations in oncogenes and tumor suppressor genes [12,13], which may impact RAC1B or SMAD3 expression independently of the cell line’s grade of differentiation or metastatic capacity. To further corroborate the association of RAC1B and SMAD3 expression with metastatic potential in an isogenic setting, we took advantage of the observation that Colo357 cells exhibit high expression of both proteins and that for this cell line, a variant (Colo357-L3.6pl) with increased metastatic potential compared to the parental cells is available. The L3.6pl (pancreas to liver) cells were generated by repeated cycles of injection of the parental cells into the pancreas of nude mice followed by retrieval of hepatic metastases and reinjection into the pancreas [19]. Strikingly, when we compared the abundance of RAC1B and SMAD3 in Colo357 and Colo357-L3.6pl cells by immunoblotting, we observed that SMAD3 was much less abundant in L3.6pl compared to the parental cells (Figure 1B). The abundance of RAC1B protein was also lower in the L3.6pl variant but differences were somewhat more variable and tightly missed statistical significance (Figure 1B).

Together, it turned out that RAC1B and SMAD3 expression at both the mRNA and protein levels is inversely correlated with metastatic capacity across several permanent PDAC cell lines and in an isogenic setting within sublines of Colo357 cells with different metastatic potential. It is therefore conceivable that RAC1B and SMAD3 lie in the same pathway and that regulatory interactions between them are involved in blocking invasion and metastasis. For the functional studies required to prove this hypothesis, we employed Panc1 and PaCa3, or BxPC3 and Capan1 cells, which are characterized by low-to-intermediate or high expression, respectively, of RAC1B and SMAD3.

### 2.2. RAC1B Knockdown or Knockout Is Associated with Decreased Abundance of SMAD3

In an attempt to reveal if coexpression of RAC1B and SMAD3 was circumstantial or if there was a regulatory link between them, we knocked down RAC1B with an exon 3b-targeting siRNA in Panc1 (Panc1–RAC1B–KD) and PaCa3 (PaCa3-RAC1B-KD) cells and assayed for SMAD3 expression by qPCR. Of note, *SMAD3* mRNA was downregulated in RAC1B siRNA-transfected Panc1 cells to 56 ± 15% (*p* < 0.038, *n* = 3, Wilcoxon test) of the respective control siRNA-transfected cells (Figure 2A). A similar effect was seen in PaCa3 cells (Appendix A). Low SMAD3 mRNA expression was also seen in Panc1 cells in which the exon 3b of *RAC1* was deleted by CRISPR/Cas technology (Panc1-RAC1B-KO cells) (19 ± 15% of control, *p* = 0.000009, *n* = 7, Wilcoxon test) (Figure 2A). In contrast, mRNA levels of the related SMAD2 remained unchanged in both RAC1B-KD and -KO cells. Immunoblot analysis revealed that Panc1-RAC1B-KO cells also exhibited a strong decrease in protein abundance of SMAD3 but not SMAD2 (Figure 2B). As expected from the low SMAD3 levels, Panc1-RAC1B-KO cells after stimulation with TGF-β1 generated less pSMAD3-C compared to vector control cells when assessed relative to the housekeeping protein, glyceraldehyde-3-phosphate dehydrogenase (GAPDH) (Figure 2C). Together, these data suggest that RAC1B promotes SMAD3 protein synthesis at the transcriptional level.

### 2.3. Knockdown of SMAD3 Mimics the Effects of Knockdown of RAC1B on Chemokinesis in Panc1 and PaCa3 Cells

We have previously shown that both Panc1-RAC1B-KD [6] and -KO [9] cells exhibit enhanced basal and TGF-β-mediated migratory activity compared to control cells. Since this was associated with reduced SMAD3 expression (see Figure 2), we hypothesized that SMAD3 is causatively involved in mediating the negative effects of RAC1B on (basal) cell motility. To test this assumption more directly and to reveal if this ability of SMAD3 is dependent on TGF-β signaling, we performed siRNA-mediated knockdown of SMAD3 followed by real-time cell migration assay in TGF-β-responsive Panc1 and PaCa3 cells, and in TGF-β-resistant BxPC3 and Capan1 cells. To this end, basal chemokinetic activity was strongly increased in Panc1-SMAD3-KD cells (Figure 3A), to a lesser extent in PaCa3-SMAD3-KD cells (Appendix A), but remained unaltered in BxPC3- and Capan1-SMAD3-KD cells, both of which exhibited very low or absent spontaneous migratory activity (data not shown) in agreement with an earlier report [14]. A quantitative comparison in Panc1 cells revealed that SMAD3 and RAC1B are equally potent in suppressing migration (Figure 3B, inset). Moreover, the combined knockdown of SMAD3 and RAC1B cells also failed to significantly increase the cells’ migratory activity over that of RAC1B or SMAD3 alone (Figure 3B, inset) and a similar trend is seen in PaCa3 cells (Appendix A). From these observations, we conclude that SMAD3’s function in TGF-β-sensitive Panc1 and PaCa3 cells is indeed antimigratory and that RAC1B and SMAD3 act through the same pathway to suppress cell migration.

### 2.4. SMAD3 Mediates the Antimigratory Effect of RAC1B

If suppression of SMAD3 is responsible for the antimigratory effect of RAC1B, then complementation of SMAD3 should be able to rescue cells from the increase in migratory activity following RAC1B knockdown. To test the prediction, we ectopically expressed SMAD3, and SMAD2 as control, in Panc1-RAC1B-KD cells and evaluated the transfectants’ basal chemokinetic activity in real-time. Intriguingly, ectopically expressed wild-type SMAD3, but not empty vector, partially prevented the increase in migration conferred by RAC1B inhibition (Figure 4A). However, in contrast to SMAD3, overexpression of SMAD2 had no effect on migration in RAC1B-KD cells (Figure 4B). These data confirm the results presented in Figure 3 and, in addition, suggest that RAC1B induces SMAD3 to inhibit cell motility in pancreatic carcinoma cells.

In agreement with the antimigratory function of RAC1B we have shown previously that individual clones of Panc1 cells stably expressing HA-tagged RAC1B (Panc1-HA-RAC1B) exhibited reduced basal migratory activity relative to empty vector controls [6]. If this reduction were mediated by upregulation of SMAD3, then knockdown of SMAD3 in Panc1-HA-RAC1B cells should prevent the decrease in migration. To test this prediction, we transfected Panc1-HA-RAC1B cells (clone 14 [6]), or empty vector-expressing control cells, with SMAD3 siRNA, or control siRNA, and subjected the transfectants to real-time cell migration assay. Intriguingly, the antimigratory effect of RAC1B was partially relieved by concomitant knockdown of SMAD3 (Figure 4C), but not SMAD2 (not shown) expression. These data reinforce the concept that suppression of random cell migration by RAC1B is mediated by SMAD3.

### 2.5. The Migration-Inhibiting Effect of SMAD3 Is Activation-Independent

Due to autocrine stimulation with TGF-β Panc1 cells may have low-level constitutive activation of SMAD signaling as evidenced by weak signals for pSMAD3-C in immunoblots of non-TGF-β-treated control cells (see Figure 2C, lanes 1, 2, 7, and 8). To analyse if this albeit small fraction of pSMAD3-C could be responsible for the antimigratory effect and to evaluate whether SMAD3 needs to be phosphorylated at the C-terminus in the first place to mediate the inhibitory effect of RAC1B on chemokinesis, we performed migration assays of Panc1-RAC1B-KD cells in the absence or presence of TGF-β1 and the SMAD3 inhibitor SIS3. This small molecule inhibits TGF-β and activin signaling by suppressing SMAD3-C phosphorylation without affecting the p38 MAPK, ERK MAPK, or PI3/AKT signaling pathways [6,21]. If the antimigratory effect of endogenous SMAD3 protein in the absence of exogenous TGF-β1 would depend on C-terminal phosphorylation, then prevention of pSMAD3-C formation by SIS3 would be expected to relieve inhibition and induce a rise in migratory activity. However, we observed an initial decrease in migration and despite slightly higher cell index values beyond the 9-h time point-these were not significantly different from those of the corresponding vehicle control (Figure 5A, compare the red and green curves). However, as predicted, migratory activity induced by TGF-β1, which strictly depends on pSMAD3-C was reduced by SIS3 treatment (Figure 5B).

To confirm these observations with a genetic approach, we ectopically expressed in Panc1 cells a C-terminal phosphorylation-resistant mutant of SMAD3 (SMAD3-mut3A) in which the three serine residues in the C-terminal SSXS motif had been converted to alanine. This mutant should combine two functional activities, the ability to inhibit C-terminal phosphorylation of SMAD3 in a dominant-negative fashion in response to TGF-β1 (like SIS3) and the ability to relieve the increase in migration upon RAC1B knockdown (same effect as wild-type SMAD3, compare with Figure 4A). As predicted, transfection of pSMAD3-mut3A partially blocked TGF-β1-induced activation of endogenous wild-type SMAD3 (Figure 5D, inset) and strongly reduced basal migration under conditions of a RAC1B knockdown (Figure 5C). In contrast, and in agreement with the SIS3 data in Figure 5B, this SMAD3 mutant dramatically inhibited TGF-β1-induced chemokinetic activity (Figure 5D). Together with the observation that SMAD3 expression did not strictly correlate with TGF-β responsiveness in a large panel of PDAC lines (see Figure 1), these data suggest that in order to antagonise basal (in contrast to TGF-β1-induced) migration, C-terminal phosphorylation of SMAD3 is dispensable. From the differential effects of SIS3 and the phosphorylation-resistant SMAD3 mutant we conclude that the antimigratory effect of SMAD3 is activation-independent.

### 2.6. Identification of Biglycan as a Common Mediator of the Antimigratory Effect of SMAD3 and RAC1B in Panc1 Cells

We have recently identified proteins that are involved in negative regulation of TGF-β1-induced cell migration, namely the cell adhesion and invasion suppressor protein E-cadherin (encoded by *CDH1*) [16] and the pericellular proteoglycan Biglycan (encoded by *BGN*) [7]. Here, we addressed the question if BGN represents a target gene of RAC1B and SMAD3. When we monitored BGN mRNA in Panc1-RAC1B-KD and PaCa3-RAC1B-KD cells it was reduced to 23.9 ± 5.4% of control (*p* < 0.001, *n* = 3, Wilcoxon test, Figure 6A, left-hand graph) and 47.5 ± 14.3% of control (*p* < 0.05, *n* = 3, Wilcoxon test, Appendix A), respectively. Conversely, in Panc1-HA-RAC1B cells, BGN mRNA abundance was increased 4.5 ± 0.9-fold over empty vector control cells (*p* < 0.01, *n* = 3, Wilcoxon test, Figure 6A, right-hand graph). We then reasoned that if BGN were responsible for the antimigratory effect of RAC1B, knockdown of *BGN* should alleviate the migration-suppressive effect of HA-RAC1B. To this end, siRNA-mediated knockdown of *BGN* in Panc1-HA-RAC1B cells almost completely reversed the antimigratory effect of HA-RAC1B (Figure 6B). Next, we analysed if *BGN* is also controlled by SMAD3 and hence if SMAD3 can mimic RAC1B’s effect on this gene. Remarkably, knocking down *SMAD3* resulted in downregulation of *BGN* in both Panc1 (Figure 6C, top left-hand graph) and PaCa3 (Appendix A) cells, whereas transient overexpression of SMAD3 or SMAD3-mut3A in Panc1 cells had the opposite effect (Figure 6C, top right-hand graph). In agreement with *BGN* being a gene that in both Panc1 and PaCa3 cells is responsive to induction by TGF-β1 in a Smad-dependent manner [22], the response of *BGN* to TGF-β1 stimulation in Panc1-SMAD3-KD (Figure 6C, bottom left-hand graph), PaCa3-SMAD3-KD (Appendix A), or in Panc1 cells overexpressing SMAD3-mut3A (Figure 6C, bottom right-hand graph) was impaired. Having shown that *BGN* was upregulated by both RAC1B and SMAD3, we speculated that knockdown of *BGN* should also be able to rescue cells from the suppressive effect of SMAD3 on chemokinesis. As shown in Figure 6D, cotransfection of Panc1 cells with a BGN siRNA but not a control siRNA was able to reverse the antimigratory activity of ectopic SMAD3 to reach almost that of vector control cells. Finally, to reveal if BGN expression is negatively associated with invasive potential, we quantified BGN mRNA in the same panel of PDAC-derived cell lines analysed in Figure 1. Intriguingly, BGN expression was significantly more abundant in poorly metastatic Colo357 and BxPC3 cells than in highly invasive PaCa3 and MiaPaCa2 cells (Appendix A). The relatively high expression of BGN in Panc1 cells (Appendix A) is likely due to high autocrine production of TGF-β1 [23] in combination with the high sensitivity of *BGN* to TGF-β induction (Figure 6C and [22]) in this cell line. These data reveal that in Panc1 and PaCa3 cells BGN is induced by RAC1B and SMAD3 and represents the common executor of their antimigratory effects.

## 3. Discussion

Prompted by the strong inhibitory effect of RAC1B on chemokinesis in pancreatic epithelial cells, we set out to elucidate the underlying mechanism and to identify potential mediator(s). In expression correlation analyses of a large panel of established PDAC lines and, in addition, of the Colo357 cell line in an isogenic background, we observed that both RAC1B and SMAD3 exhibit a very similar expression pattern that correlates positively with the cell lines’ differentiation grade and negatively with their metastatic potential. With respect to SMAD3, this was somehow counterintuitive as three out of four SMAD3^high^ PDAC lines lack SMAD4 expression and, as a consequence, have lost the ability to respond to TGF-β stimulation with activation of this pathway [10,11,12,13]. Nevertheless, our data suggested the possibility that SMAD3 has cellular function(s) independent of its C-terminally phosphorylated form and that RAC1B and SMAD3 are functionally linked. It should also be mentioned that the greater abundance of RAC1B expression in cell lines with low metastatic potential is in good agreement with earlier data in PDAC tissues in situ where RAC1B overexpression in the tumor cell fraction was associated with prolonged overall survival [6].

Using RNA interference and CRISPR/Cas-mediated genomic deletion of exon 3b of *RAC1* in Panc1 cells we observed that knockdown or knockout of RAC1B was associated with a decrease in both mRNA and protein abundance of SMAD3 but not SMAD2. Upon treatment with TGF-β1, the levels of pSMAD3-C in Panc1-RAC1B-KO cells were lower than in the empty vector control cells, however, despite lower abundance of both (TGF-β1-induced) pSMAD3-C and total SMAD3, their ratio (pSMAD3-C/SMAD3) in the RAC1B depleted cells still exceeded that in the vector control cells [9].

Given the positive association of RAC1B with SMAD3 expression, we addressed the question of whether there is a functional link between both proteins that impacts cell motility. Previous studies have shown that RAC1B negatively regulates TGF-β signaling in pancreatic cells by downregulating the expression of several positive mediators and upregulating that of inhibitory factors of the pathway, thereby blocking more complex responses to this growth factor such as growth inhibition, EMT, and cell migration [7,9]. Central to this phenomenon is the suppression of the abundance and activity of ALK5 [9]. Based on RAC1B’s ability to control gene expression in a way that favours silencing of TGF-β pathway activity, we considered the possibility that SMAD3 has an inhibitory function independent of its role as a TGF-β signal transducer. To this end, we show that RAC1B utilizes SMAD3 protein to suppress random cell migration in the absence of any exogenously added stimulators. This is inferred from the observations that (i) ectopic expression of SMAD3 alone mimicked the antimigratory effect of RAC1B, and (ii) both ectopic overexpression or knockdown of SMAD3 rescued the cells from the RAC1B knockdown-induced increase or overexpression-induced decrease, respectively, in cell migration. The observations that the effects of knocking down RAC1B or SMAD3 did not differ significantly and that a combined knockdown of SMAD3 and RAC1B was neither additive nor synergistic, confirmed our contention that RAC1B and SMAD3 operate through the same pathway to control cell migration.

Our findings suggest the possibility that (non-C-terminally phosphorylated) SMAD3 has functions distinct from its phosphorylated/activated form(s) (Figure 7). This scenario is supported by the observations that SMAD3 suppresses the migratory activity in TGF-β-sensitive Panc1 and PaCa3 in the absence of exogenous TGF-β1 in both its wild-type form and as a C-terminal phosphorylation-resistant mutant. However, at present, we cannot exclude a role for non-C-terminal phosphorylation of SMAD3 in conveying the antimigratory effect, since both SMAD2 and SMAD3 can also be phosphorylated in their linker regions by MAPKs and members of the nuclear cyclin-dependent kinase (CDK) family [24]. Intriguingly, knockdown of *SMAD3* in TGF-β-resistant BxPC3 and Capan1 cells failed to increase migration although any weak effect may have been obscured by their low basal migratory activity [14], which in turn may be due to low expression of vimentin and high expression of the invasion suppressor, E-cadherin [16]. Panc1 and PaCa3 cells differ from BxPC3 and Capan1 cells, not only by their strong invasive potential and functional TGF-β signaling, but also by high autocrine TGF-β1 production in vitro [10,11,23]. In Panc1 and other TGF-β-sensitive PDAC lines, TGF-β has been shown to upregulate its own expression and to act in an autocrine manner to drive cell migration and invasion in vitro in a cell-autonomous manner, while in cell lines with deficient SMAD4 expression (BxPC3, Capan1, Capan2), TGF-β failed to autoinduce its own expression [10,11]. Although more cell lines of low and high TGF-β sensitivity and autocrine signaling need to be studied, nevertheless, our data suggest that the ability of SMAD3 to inhibit migration is dependent on functional TGF-β/SMAD signaling. In addition, whether autocrine TGF-β stimulation is required for SMAD3 to suppress migration may be studied in Colo357 cells, which are highly TGF-β-sensitive, but exhibit only a small amount of TGF-β autocrine expression [23].

The observation of this dual role of SMAD3 reconciles with previous reports that SMAD3 can regulate target genes through a TGF-β-independent pathway [25,26,27]. Although SMAD3 when acting downstream of TGF-β or activin receptors promotes the morphological changes that characterize EMT through the transcriptional regulation of target genes in tumorigenesis and fibrosis of the eye or kidney [28,29,30], evidence exists indicating that SMAD proteins, and particularly SMAD3, may also be able to inhibit or even reverse EMT, cell motility and metastasis [31,32,33]. For instance, Pohl and colleagues showed that the restoration of expression of SMAD4 (the binding partner of SMAD3 required in order for SMAD3 to enter the nucleus and activate its target genes) in SW480 colon carcinoma cells led to reversion from a mesenchymal-like phenotype to an epithelial-like phenotype [34].

An interesting issue relates to the molecular mechanism of SMAD3 regulation by RAC1B. It remains to be determined whether RAC1B stimulates SMAD3 mRNA levels via an increase in promoter activity or mRNA stability, or via downregulation of a SMAD3 targeting microRNA. Another issue worth analysing is whether SMAD3 regulation by RAC1B is direct or whether it involves the expression or activation of an intermittent factor(s). We have preliminary data to indicate that RAC1B can induce SMAD3 through production of autocrine TGF-β1 and itself is negatively regulated by TGF-β1, thus forming a negative feedback loop. This observation could help to explain why BxPC3 and Capan1 cells despite defective Smad signaling exhibit strong expression of RAC1B and SMAD3. Furthermore, we would like to speculate that the antimigratory effect of SMAD3 is confined to cells with functional TGF-β signaling and possibly autocrine TGF-β production because only in these cells BGN expression is upregulated to sufficiently high levels to be able to block migration [22]. Downstream of RAC1B and SMAD3, we were able to identify BGN as an executor of the antimigratory effect of this pathway in Panc1 and PaCa3 cells consistent with known antimigratory function of this proteoglycan [7,35]. Based on these findings, we propose a model with dual functionality of how RAC1B negatively controls cell migration: prevention of pSMAD3-C formation and thus TGF-β/Smad-dependent induction of migration-associated gene expression through direct downregulation of ALK5 [9] (Figure 7, left-hand side), and indirectly through (non-C-terminally phosphorylated) SMAD3-mediated induction of *BGN* (Figure 7, right-hand side). The migration-suppressive effect of BGN may involve a reduction in the bioavailability of TGF-β for its receptors [36] (Figure 7, centre). Interestingly, BGN has been shown to be overexpressed in human PDAC [37] and to mediate the anti-invasive and antimetastatic function of the tumor suppressor protein TAp73 in a genetic mouse model of PDAC [35]. Upon silencing of TAp73 in a cell line derived from these mice, both Bgn and Smad3 were downregulated, suggesting the exciting possibility that Rac1b is targeted by TAp73 to participate in its tumor-suppressive function in pancreatic cancer. We are currently evaluating whether blockage of cell motility in pancreatic epithelial cells is the result of more general role of RAC1B and SMAD3 in maintaining a differentiated epithelial phenotype and in protecting the cells from undergoing mesenchymal transdifferentiation.

It remains to be seen whether RAC1B (over)expression in pancreatic tumors correlates positively with SMAD3 expression in vivo. The SMAD3 phosphoisoform expression is being used as a surrogate marker for TGF-β signaling activity in tumor tissue in situ and has prognostic significance [38,39]. Given the potent antimigratory effect of SMAD3 induced by RAC1B, or possibly also other factors, it may be worth to separately evaluate pSMAD3-C and total SMAD3 expression in tumors. This will reveal if (non-C-terminally phosphorylated) SMAD3 represents an independent prognostic factor, whose overexpression will in contrast to pSMAD3-C predict a better outcome. In sum, this study has identified *SMAD3* and *BGN* as novel RAC1B target genes in pancreatic cancer cells with SMAD3, in an activation-independent manner, inducing BGN, which subsequently executes inhibition of cell motility in TGF-β-dependent and/or independent fashion.

## 4. Material and Methods

### 4.1. Antibodies and Reagents

The following primary antibodies were used: anti-Smad3, #ab40854, Abcam (Cambridge, UK), anti-Smad2, #sc-7947, anti-HSP90, #sc-13119, and anti-TGF-β receptor I, V22, #sc-398, Santa Cruz Biotechnology (Heidelberg, Germany), anti-RAC1B, #09-271, Merck Millipore (Darmstadt, Germany), and anti-phospho-Smad3(Ser423/425), #9514, and anti-GAPDH (14C10), #2118, Cell Signaling Technology (Frankfurt am Main, Germany). HRP-linked anti-rabbit, #7074, and anti-mouse, #7076, secondary antibodies were from Cell Signaling Technology. Recombinant human TGF-β1, #300-023, was provided by ReliaTech (Wolfenbüttel, Germany), and SIS3 from Merck/Calbiochem. The C-terminal phosphorylation-resistant SMAD3 mutant (SMAD3-mut3A), in which the three serine residues at position 422, 423, and 425 were replaced by alanines, was constructed by amplifying the entire coding sequence of SMAD3 with forward primer 5’-ACCATGGCGTCCATCCTGCCTTTC-3’ (Kozak consensus sequence underlined) and reverse primer 5’-TCTAAGCCACAGCGGCACAGCGGATGCTTGGGG-3’ (mutant nucleotides underlined) with *Pfu* polymerase and subsequent insertion in pcDNA3 digested with *Eco*RV. The desired sequence changes were confirmed by sequencing. 

### 4.2. Cells

Panc1 cells derived from human PDAC were obtained from the American Type Culture Collection (ATCC) (Manassas, VA, USA). The other PDAC cell lines were supplied by H. Kalthoff (Kiel, Germany) except for Colo357-L3.6pl cells, which were a gift from Dr. U. Wellner (Lübeck, Germany). All cell lines were maintained in RPMI 1640 supplemented with fetal bovine serum (FBS), 1% penicillin-streptomycin-glutamine (Life Technologies, Darmstadt, Germany) and 1% sodium pyruvate (Merck Millipore). The generation and characterization of Panc1 cells stably expressing HA-RAC1B or engineered by CRISPR/Cas9 technology to lack exon 3b of *RAC1* has been described in detail earlier [6,9].

### 4.3. Transient Transfection of siRNA and Expression Vectors

On day 1, cells were seeded into Nunclon^TM^ Delta Surface plates (Nunc, Roskilde, Denmark), and transfected twice, on days 2 and 3, serum-free with either 25 or 50 nM of prevalidated siRNAs specific for RAC1B [6], SMAD2, SMAD3 (both from Dharmacon, Lafayette, CO, USA), BGN [7], or the respective scrambled controls, and/or pcDNA3-based expression vectors for SMAD3, SMAD3-mut3A, or SMAD2, for 4 h using Lipofectamine 2000 (Life Technologies). Additional validation of these siRNAs was performed for RAC1B [6] and SMAD2/SMAD3 (this study).

### 4.4. QRT-PCR Analysis

Total RNA was extracted from Panc1 cells using PeqGold RNAPure from Peqlab (Erlangen, Germany) and purified according to manufacturer’s instructions. For each sample, 2.5 μg RNA were subjected to reverse transcription for 1 h at 37 °C, using 200 U M-MLV Reverse Transcriptase and 2.5 μM random hexamers (Life Technologies) in a total volume of 20 μL. Relative mRNA expression of target genes was quantified by qPCR on an I-Cycler (BioRad, Munich, Germany) using Maxima SYBR Green Mastermix (Thermo Fisher Scientific, Waltham, MA, USA). Data were normalized to the expression of GAPDH. For sequences of PCR primers, see Appendix A.

### 4.5. Immunoblotting

Cell lysis and immoblotting was essentially performed as described previously [22]. In brief, cells were washed once with ice-cold PBS and lysed with 1× PhosphoSafe lysis buffer (Merck Millipore). Following clearance of the lysates by centrifugation, their total protein concentrations were determined with the DC Protein Assay (BioRad). Equal amounts of proteins were fractionated by polyacrylamide gel electrophoresis on mini-PROTEAN TGX any-kD precast gels, or TGX Stain-Free FastCast gels (BioRad) and blotted to polyvinylidene fluoride (PVDF) membranes. Membranes were blocked with nonfat dry milk or bovine serum albumin and incubated with primary antibodies either for 2 h at RT or overnight at 4 °C. After washing and incubation with HRP-linked secondary antibodies, chemoluminescent detection of proteins was done on a ChemiDoc XRS+ System with Image Lab Software (BioRad) with Amersham ECL Prime Detection Reagent (GE Healthcare, Munich, Germany). The signals for the proteins of interest were normalized to either the total amount of protein in the same lane (when using the TGX Stain-Free FastCast gels), or to bands for the housekeeping gene GAPDH.

### 4.6. Migration Assays

We employed the xCELLigence^®^ DP system (ACEA Biosciences, San Diego, distributed by OLS, Bremen, Germany) to measure chemokinesis of Panc1 and MDA-MB-231 cells. Briefly, CIM plates-16 were prepared according to the instruction manual and previous descriptions [6,7]. The underside of the upper chambers of the CIM plate-16 was coated with 30 μL of collagen I to facilitate adherence of the cells and enhance signal intensities. After filling the wells of the lower chambers with 170 µL of normal growth medium containing 1% FBS, the lower and upper chambers of the CIM plate-16 were assembled, the wells of the upper chamber supplied with 50 µL of the same medium and the whole device equilibrated in the incubator for 1 h. The upper chamber of each well was then loaded with 50,000–100,000 cells suspended in 100 µL of RPMI with 1% FBS. In some experiments, TGF-β1 was added to the cell suspensions at a concentration of 5 ng/mL. Data acquisition was done at intervals of 15 min and analysed with real-time cell analysis (RTCA) software, version 1.2 (ACEA) and displayed as the dimension-less cell index (CI) reflecting relative migratory activity. Quantification of migration was done by determining the cell CI at the time point of peak migratory activity and subtracting from this the corresponding CI of the control cells [9].

### 4.7. Statistical Analysis

Statistical significance was calculated using the Wilcoxon test or the Mann–Whitney *U*-test. Results were considered significant at *p* < 0.05 (*). Higher levels of significance were *p* < 0.01 (**) and *p* < 0.001 (***).

## 5. Conclusions

The data of this study report on an unexpected role of (non-C-terminally phosphorylated) SMAD3 in transducing an antimigratory signal derived from RAC1B in pancreatic carcinoma cells. Upregulation of SMAD3 and the small proteoglycan BGN is likely to be one route through which this RAC1 isoform may block invasion and metastasis. Therefore, promoting RAC1B generation from *RAC1* and/or *SMAD3* and *BGN* expression may be a novel and promising antiinvasive/antimetastatic strategy in PDAC therapy.

## Figures and Tables

**Figure 1 cancers-11-01959-f001:**
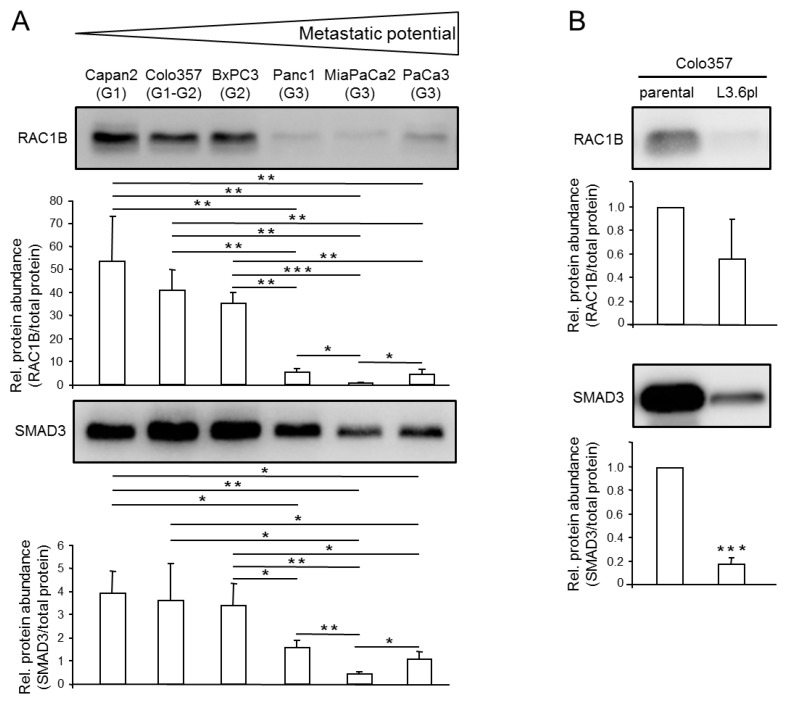
Expression of Ras-related C3 botulinum toxin substrate 1B (RAC1B) and Mothers against decapentaplegic homolog 3 (SMAD3) in pancreatic ductal adenocarcinoma (PDAC)-derived cell lines of different metastatic potential. (**A**) Basal protein expression of RAC1B and SMAD3 in the indicated permanent PDAC cell lines (the differentiation grade, G1, G2, or G3, is given in brackets). Cells were grown to approximately 80% confluence before preparation of cell lysates and processing for immunoblotting with antibodies to RAC1B (upper blot) or SMAD3 (lower blot). Shown are representative blots. The graphs underneath the blots show results from densitometric analyses of RAC1B and SMAD3 band intensities after normalization to the amount of total protein in the same lane. Data represent the combined results of four samples for each cell line (mean ± SD, *n* = 4). (**B**) Immunoblot analysis of RAC1B and SMAD3 in parental Colo357 cells (parental) and Colo357-L3.6pl cells (L3.6pl). Data quantification was performed as described in (**A**). The asterisks (* *p* < 0.05, ** *p* < 0.01, *** *p* < 0.001) indicate significant differences (Wilcoxon test). The uncropped blots and molecular weight markers are shown in Appendix A.

**Figure 2 cancers-11-01959-f002:**
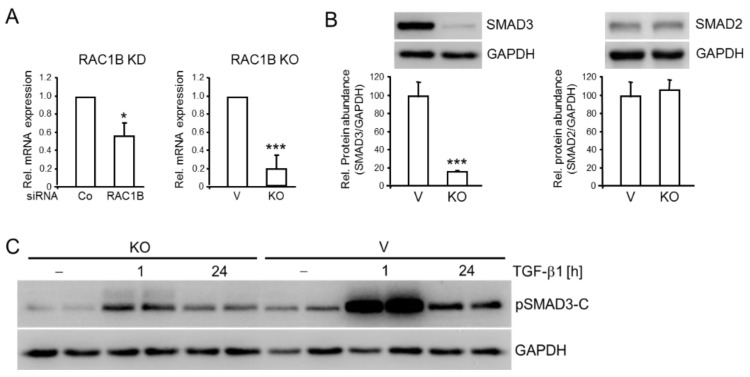
RAC1B depletion is associated with downregulation of SMAD3 expression in Panc1 cells. (**A**) mRNA expression of SMAD3 in Panc1-RAC1B-KD (left graph) and -KO cells (right graph) as determined by qPCR analysis. Panc1-RAC1B-KD cells were generated by transient transfection with 50 nM of either irrelevant control siRNA (Co) or RAC1B siRNA as outlined in the Methods section. Shown are the means ± SD calculated from seven experiments. V, vector control. (**B**) Immunoblot detection of total SMAD3 (left graph) or SMAD2 (right graph) protein in lysates of Panc1-RAC1B-KO and vector control cells. The graph below the blots shows results from densitometry-based quantification of band intensities (means ± SD from six experiments). (**C**) Panc1-RAC1B-KO cells and vector controls were stimulated with TGF-β1 for various times, as indicated, and duplicate samples for each time point were subjected to immunoblot analysis of pSMAD3-C, and glyceraldehyde-3-phosphate dehydrogenase (GAPDH) as loading control. The asterisks (* *p* < 0.05, *** *p* < 0.001) in (**A**,**B**) indicate significance (Wilcoxon test). The uncropped blots and molecular weight markers are shown in Appendix A.

**Figure 3 cancers-11-01959-f003:**
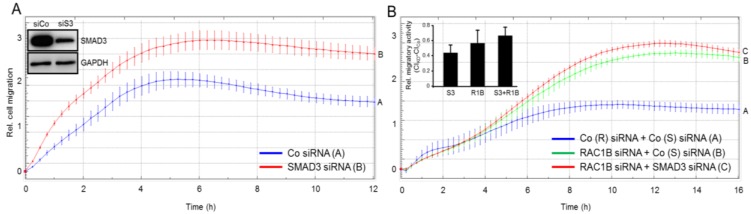
Effect of SMAD3 knockdown on basal migratory activity in Panc1 cells. (**A**) Panc1 cells were transiently transfected twice with 50 nM of either control (Co) siRNA or SMAD3 siRNA. Forty-eight h after the second round of transfection, cells were subjected to real-time cell migration assay. Inset, functional validation of the SMAD3 siRNA by immunoblotting. siCo, Co siRNA; siS3, SMAD3 siRNA. (**B**) Panc1 cells were transiently transfected twice with 25 nM each of Co siRNA for SMAD3 + Co siRNA for RAC1B, RAC1B siRNA + Co siRNA for SMAD3, or RAC1B siRNA + SMAD3 siRNA. Forty-eight h after the second round of transfection cells were subjected to real-time cell migration assay as in (**A**). Inset, Quantitative comparison of SMAD3 and RAC1B knockdown on migratory activities. Data shown are the mean ± SD (*n* = 4). Differences in the mean values among the three treatments did not reach statistical significance. Data in (**A**,**B**) are from one representative experiment out of four experiments with very similar results, and are the mean ± SD from 3–4 wells per condition. In (**A**), differences between Panc1 + SMAD3 siRNA (red curve, tracing B) and Panc1 + Co siRNA (blue curve, tracing A) are significant at 0:30 and all later time points (*p* < 0.05, Mann-Whitney *U*-test). In (**B**), differences between Panc1 + RAC1B siRNA + SMAD3 siRNA (red curve, tracing C) and Panc1 + Co siRNAs (blue curve, tracing A) are significant at 5:30 and all later time points (*p* < 0.05, Mann-Whitney *U*-test).

**Figure 4 cancers-11-01959-f004:**
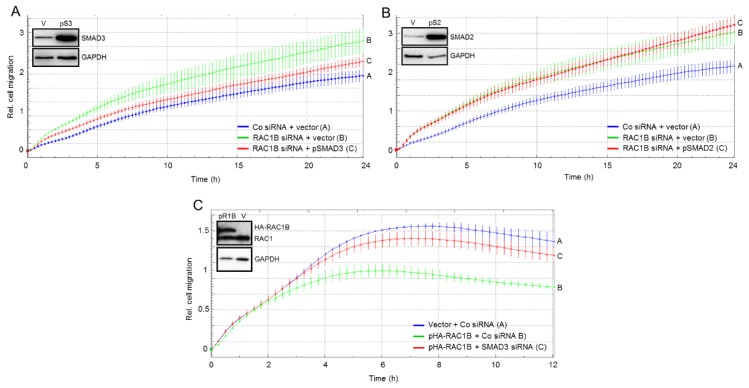
Effect on random cell migration of ectopic expression of SMAD3 or SMAD2 in Panc1-RAC1B-KD cells or SMAD3 knockdown in Panc1-HA-RAC1B cells. Panc1 cells were transiently transfected twice with 50 nM of either Co or RAC1B siRNA and in the second transfection received, in addition, 4 μg/mL of empty vector (pcDNA3, V), or either pSMAD3 (pS3) (**A**), or pSMAD2 (pS2) (**B**). Forty-eight h after the second round of transfection cells were subjected to real-time cell migration assay for 24 h. The insets show successful overexpression of SMAD3 and SMAD2 at the time of assay initiation. In both panels the experiment shown is representative of three independent assays and depicts the means ± SD from 3–4 parallel wells. In (**A**) differences between Panc1 + RAC1B siRNA + pSMAD3 (red curve, tracing C) and Panc1 + RAC1B siRNA + vector (green curve, tracing B) are significant at 4:30 and all later time points (*p* < 0.05, Mann-Whitney *U*-test). (**C**) Effect of SMAD3 knockdown on random cell migration in Panc1 cells with stable ectopic expression of RAC1B. Panc1-HA-RAC1B cells (pR1B, clone 14) or Panc1 cells transduced with empty pCGN vector (vector, V) were transiently transfected twice with 50 nM of either Co siRNA or SMAD3 siRNA. Forty-eight h after the second round of transfection cells were subjected to real-time cell migration assay. Data are the means ± SD from 3–4 parallel wells taken from a representative experiment (three performed in total). Differences between Panc1-HA-RAC1B cells + SMAD3 siRNA (red curve, tracing C) and Panc1-HA-RAC1B cells + Co siRNA (green curve, tracing B) are significant at 3:30 and all later time points (*p* < 0.05, Mann–Whitney *U*-test). The insets show immunoblots with successful overexpression of the transfected proteins.

**Figure 5 cancers-11-01959-f005:**
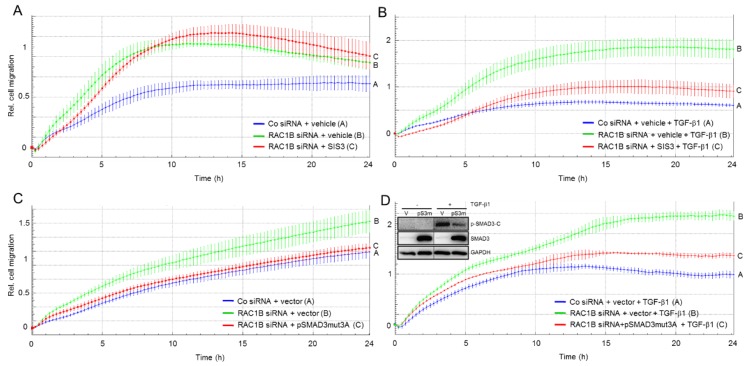
The effect of inhibitors of SMAD3 activation on basal and TGF-β1-induced cell migration. (**A**) Panc1 cells were transiently transfected twice with 50 nM of either Co siRNA or RAC1B siRNA and subsequently subjected to migration assay in the presence of either vehicle (0.1% dimethylsulfoxide) or 10 μM SIS3. (**B**) As in (**A**), except that all cells received, in addition, TGF-β1. (**C**) As in (**A**), except that in addition to RAC1B siRNA Panc1 cells were transfected with either the C-terminal phosphorylation-resistant pSMAD3-mut3A or empty pcDNA3 vector (vector) instead of treatment with SIS3. (**D**) As in (**C**), except that the cells were stimulated with TGF-β1. The assays in (**A**–**D**) are each representative of three experiments and depict the means ± SD from 3–4 parallel wells. In (**B**) differences between RAC1B siRNA-transfected and SIS3 + TGF-β-treated cells (red curve, tracing C) and RAC1B siRNA-transfected and vehicle + TGF-β-treated cells (green curve, tracing B) are significant at 1:30 and all later time points (*p* < 0.05, Mann-Whitney *U*-test). In (**C**) differences between RAC1B siRNA + pSMAD3-mut3A-transfected cells (red curve, tracing C) and RAC1B siRNA + vector-transfected cells (green curve, tracing B) are significant at 3:00 and all later time points (*p* < 0.05, Mann–Whitney *U*-test). In (**D**) differences between RAC1B siRNA + pSMAD3-mut3A-transfected and TGF-β1-treated cells (red curve, tracing C) and RAC1B siRNA + vector-transfected TGF-β-treated cells (green curve, tracing B) are significant at 5:00 and all later time points (*p* < 0.05, Mann–Whitney *U*-test). Inset, the functionality of pSMAD3-mut3A and SIS3 in blocking the formation of pSMAD3-C after a 1-h treatment with TGF-β1 was verified by immunoblotting.

**Figure 6 cancers-11-01959-f006:**
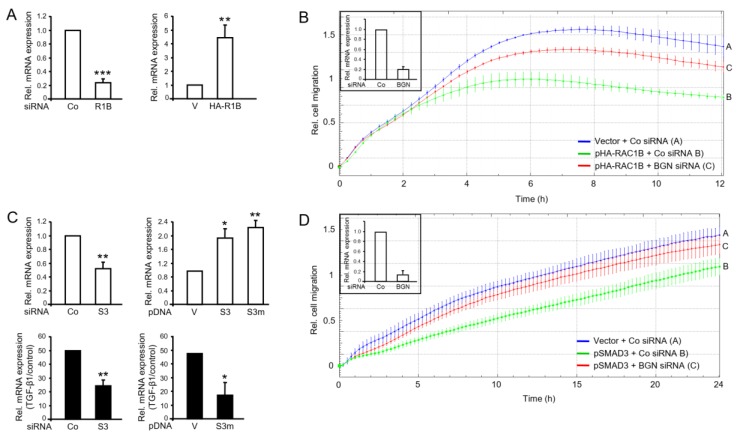
The small proteoglycan biglycan (BGN) mediates the inhibitory effects of RAC1B and SMAD3 on chemokinesis in Panc1 cells. (**A**) BGN expression was assayed by qPCR analysis in Panc1 cells transiently transfected with 50 nM of either Co siRNA or RAC1B (R1B) siRNA (top left-hand graph), and in Panc1-HA-RAC1B cells (HA-R1B) or empty vector (V) control cells (top right-hand graph). Data are the mean ± SD (*n* = 3). (**B**) Real-time cell migration assay of Panc1-HA-RAC1B cells (pHA-RAC1B) or vector control cells (Vector) with either Co or BGN siRNA as indicated. Data are the mean of 3 parallel wells (mean ± SD). One representative assay out of three performed in total is shown. Differences between Panc1-HA-RAC1B + BGN siRNA (red curve, tracing C) and Panc1-HA-RAC1B + Co siRNA (green curve, tracing B) are significant at 3:30 and all later time points (*p* < 0.05, Mann-Whitney *U*-test). Successful inhibition of BGN expression was verified by qPCR (inset). (**C**) Panc1 cells were transiently transfected twice with 50 nM of either Co siRNA or SMAD3 (S3) siRNA (top left-hand graph), or plasmid DNA (pDNA), empty vector (V), pSMAD3 (S3), or pSMAD3-mut3A (S3m) (top right-hand graph), and 48 h after the second transfection analysed for expression of *BGN* by qPCR. A fraction of all transfectants was treated with TGF-β1 for 24 h prior to the BGN qPCR (bottom graphs). Data represent the mean ± SD of three transfections. (**D**) Real-time cell migration assays of Panc1 cells cotransfected with empty vector (Vector) or pSMAD3 in combination with either Co or BGN siRNA as indicated. Data are the mean of 3 parallel wells (mean ± SD). The assay shown is representative of three assays performed in total. Differences between Panc1 + pSMAD3 + BGN siRNA (red curve, tracing C) and Panc1 + pSMAD3 + Co siRNA (green curve, tracing B) are significant at 3:30 and all later time points (*p* < 0.05, Mann-Whitney *U*-test). Successful inhibition of BGN expression was verified by qPCR (inset). The asterisks (* *p* < 0.05, ** *p* < 0.01, *** *p* < 0.001) in (**A**,**C**) indicate significance (Wilcoxon test).

**Figure 7 cancers-11-01959-f007:**
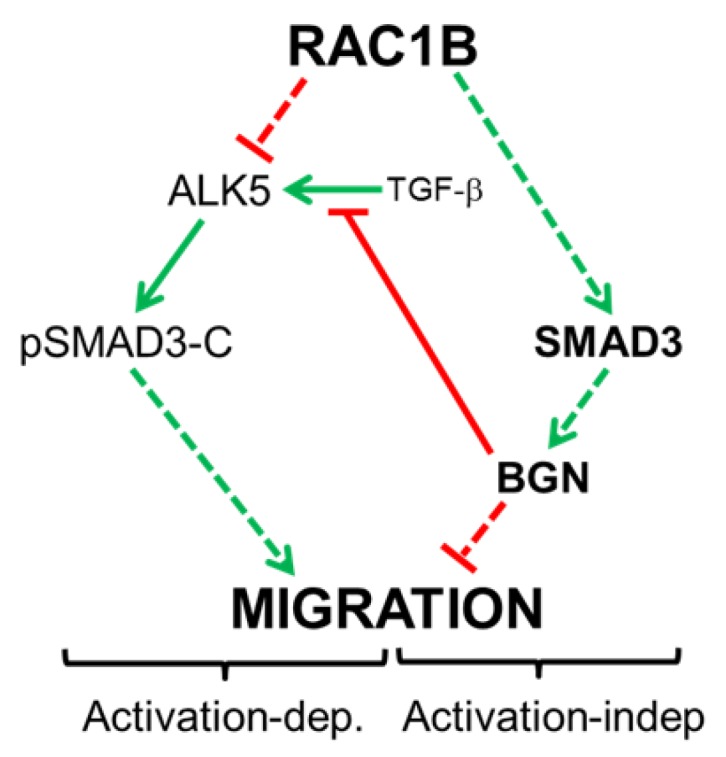
Diagram illustrating the regulatory interactions between RAC1B and SMAD3 and the dual functionality of SMAD3 with respect to control of cell migration in highly invasive pancreatic cancer cells with functional TGF-β signaling. RAC1B blocks formation of pSMAD3-C in response to TGF-β stimulation by suppressing expression of the TGF-β type I receptor ALK5. This in turn prevents TGF-β1-induced activation-dependent (Activation-dep.) migration (left-hand side). At the same time, RAC1B stimulates the expression of (non-C-terminally phosphorylated) SMAD3, which in turn inhibits migration in an activation-independent (Activation-indep.) manner by promoting the expression of BGN (right-hand side). However, both routes are interconnected through BGN, which can bind TGF-β extracellularly, thereby preventing access to its receptor, ALK5 (center). Stimulatory interactions are indicated by green arrows and inhibitory interactions by red lines. Solid lines indicate direct interactions, whereas stippled lines indicate the possibility that interactions involve intermediary proteins.

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
