# Peer review of "Negative Control of Cell Migration by Rac1b in Highly Metastatic Pancreatic Cancer Cells Is Mediated by Sequential Induction of Nonactivated Smad3 and Biglycan"

_cancers, 2019, doi:10.3390/cancers11121959_

Round 1

Reviewer 1 Report

The authors have studied the role of RAC1B and SMAD3 in cell migration. They conclude that both proteins and Byglycan expression control, in a negative way, cell migration.  The manuscript is well written and the results are interesting. However, some aspects expressed below, have been taking into account to recommend major revision.

One of the main recommendations is that the author should perform the functional experiments in more than one cancer cells (not only in Panc1 cell line). Actually, the journal does not accept research articles using only one cell line for the experiments. The author should consider using different cell lines, also with different metastatic potentials to demonstrate if the effect of silencing RAC1B or SMAD3 or the use of the inhibitor is correlated with the migratory effect.

In line 90: “Given the known anti-migratory role of RAC1B in Panc1 cells, we reasoned that RAC1B might 91 be of more general importance in determining the migratory/invasive potential of PDAC-derived 92 cells and hence might be enriched in highly metastatic cells.” I think it should be poorly metastatic cells. In some other sentences, the use of the term “correlated” may be accompanied by “negatively” of “positively”.

Could the author better explain how SMAD3 was selected to further investigate? They found a parallel level of expressions but it is not clear how they established the connection between both proteins to study the migration.

The authors state that the panel of cells studied had different levels of sensitivity to TGF-β. Could they show the experiments done to have these conclusions? Have they measured the migration, cell scattering, expression of receptors, epithelial-mesenchymal transdifferentiation? As the authors conclude that RAC1B and SMAD3 induce an anti-migratory effect independent of TGF-β-treatment (Figure 7), would be interested to perform the migration experiments (figures 3 to 6) in high and low sensitive cell lines to TGF-b.

Is there any insight how RAC1B induces the expression of SMAD3?

Could the authors compare which siRNA-mediated inhibition: RAC1B or SMAD3 or both together is more effective to reverse the anti-migratory effects?

In line 196, authors have written that the performed the experiment in two cell lines, however, only experiments performed in Panc-1 are shown.

The authors should reconsider to change this title “Ectopic Expression of SMAD3 Rescues Panc1 Cells from the RAC1B Knockdown-Induced Increase in Migratory Activity whereas Knockdown of SMAD3 Rescues Panc1 Cells from the RAC1B Overexpression-Induced Decrease in Migratory Activity” is long and difficult to understand.

How the authors can explain the differences found between the SIS3 drug and the pSMAD3-mut3A (Figures 5A and 5C)? If I well understood, in both cases there is an inhibition of the phosphorylation of the C-terminal of the protein. However, both approaches have the same effect after TGF-b treatment. This explanation is not clear “these data suggest that in order to antagonize basal migration, C-terminal phosphorylation of SMAD3 is dispensable. However, SMAD3-C phosphorylation is nevertheless required to promote cell migration in response to TGF-b1 stimulation.”

In vivo assays in immunocompetent mice would be very informative to determine the role of the axis RAC1B/SMAD3/BGN in cell migration and metastasis. Besides the anti-migratory effect that the author shown in the manuscript, previously it has been shown that Smad3 has different roles in the tumor development, inducing immunosuppression or EMT. An in vivo model could clarify if promoting RAC1B

and inducing SMAD3 and BGN expression may be a therapeutic strategy for PDAC.

Author Response

The authors have studied the role of RAC1B and SMAD3 in cell migration. They conclude that both proteins and Byglycan expression control, in a negative way, cell migration.  The manuscript is well written and the results are interesting. However, some aspects expressed below, have been taking into account to recommend major revision.

One of the main recommendations is that the author should perform the functional experiments in more than one cancer cells (not only in Panc1 cell line). Actually, the journal does not accept research articles using only one cell line for the experiments. The author should consider using different cell lines, also with different metastatic potentials to demonstrate if the effect of silencing RAC1B or SMAD3 or the use of the inhibitor is correlated with the migratory effect.

Response: We thank the reviewer for this suggestion. Indeed, we included migration data with the breast cancer cell line MDA-MB-231 which showed that SMAD3 knockdown enhanced  migration (see original Suppl. Fig. 2). However, we agree that it is reasonable to compare the data in Panc1 cells with those in other PDAC-derived cell lines. We have therefore performed migration assays with poorly metastatic BxPC3 and Capan1 and highly metastatic PaCa3 cells following knockout of RAC1B or SMAD3. Unfortunately and not unexpectedly, the basal migration in BxPC3 and Capan1 cells was too low in our impedance-based real-time assays to be able to measure statistically significant differences between the various transfectants. In both cell lines there was a trend to enhanced migratory activity after RAC1B but not SMAD3 knockdown. In contrast, in PaCa3 cells we observed enhanced basal migration after knockdown of both RAC1B and SMAD3. These data are shown in Supplementary Figure 3 in the revised version. Interestingly, the ability to migrate in the absence of TGF-β appears nevertheless to be associated with a functional TGF-β pathway and/or the ability for autocrine production/secretion of TGF-b. See point #4.

In line 90: “Given the known anti-migratory role of RAC1B in Panc1 cells, we reasoned that RAC1B might be of more general importance in determining the migratory/invasive potential of PDAC-derived cells and hence might be enriched in highly metastatic cells.” I think it should be poorly metastatic cells. In some other sentences, the use of the term “correlated” may be accompanied by “negatively” of “positively”.

Response: This is correct. Both issues have been rectified.

Could the author better explain how SMAD3 was selected to further investigate? They found a parallel level of expressions but it is not clear how they established the connection between both proteins to study the migration.

Response: For quite some time our research interests focussed on the crosstalk of RAC1B with TGF-β signaling in the regulation of cell motility with the goal to identifiy TGF-β signaling intermediates that are targeted by RAC1B. When we analysed Panc1-RAC1B-knockout cells for C-terminal phosphorylation of SMAD3 we also probed the blots for total SMAD3 protein and observed - much to our surprise - that it was much less abundant in the knockout compared to the control cells. Since this contrasted with the concomitant increase in basal migratory activity of the RAC1B knockout cells (see Ref. #9) we pursued the idea that (non-phosphorylated) SMAD3 may be a mediator of RAC1B’s antimigratory effect.

The authors state that the panel of cells studied had different levels of sensitivity to TGF-β. Could they show the experiments done to have these conclusions? Have they measured the migration, cell scattering, expression of receptors, epithelial-mesenchymal transdifferentiation? As the authors conclude that RAC1B and SMAD3 induce an anti-migratory effect independent of TGF-β-treatment (Figure 7), would be interested to perform the migration experiments (figures 3 to 6) in high and low sensitive cell lines to TGF-b.

Response: Data on the TGF-β sensitivity of the various cell lines were mostly derived from published studies. Insensitivity to TGF-b is the result of mutational inactivation or lack of expression of genes critically involved in TGF-b signaling (see Refs. #12 and #13 in the revised version and Grau et al. 1997, Cancer Res 57:3929). The most commonly mutated gene in PDAC (~50% of all tumors) is DPC4 which encodes SMAD4. SMAD4 protein is absent from BxPC3 (homozygous deletion of DPC4), Capan1 (point mutation with loss of expression), and Capan2 cells (loss of expression, Ref. #11), while MiaPaCa2 cells lack expression of the TGF-b type II receptor (Ref. #15). As a consequence of these alterations, BxPC3, Capan1 and MiaPaCa2 were found to be resistant to many TGF-b responses, i.e. growth inhibition (Grau et al. 1997, Cancer Res 57:3929).

Among the cell lines of our panel only Colo357, Panc1 (Refs. #6, #7, and #10), and PaCa3 (this study) cells have retained sensitivity to TGF-b. The rationale for using both TGF-b sensitive and resistant cells has been explained in two sentences added to the last paragraph of the Introduction. The ratio of TGF-b-sensitive to resistant cell lines actually reflects the in vivo situation where 50-60% of tumors are TGF-b signaling defective.

As mentioned above, we have performed the migration experiments in both TGF-b-sensitive and resistant cell lines but basal migratory activities in transfected BxPC3 and Capan1 cells (both poorly metastatic) were too low to be able to safely assess the effects of RAC1B and SMAD3 knockdown. Knockdown of RAC1B appeared to enhance migration but knockout of SMAD3 did not. In contrast, Panc1, PaCa3 (and MDA-MB-231 breast cancer cells) cells responded with increased basal migration following knockdown of SMAD3 or RAC1B. The new data with PaCa3 cells have been included in the revised version as part of the Supplementary Material (i.e. real-time migration assay in Fig. S3). It should also be mentioned that the use of MiaPaCa2 cells in the knockdown/migration experiments was not feasible due to extremely low expression of RAC1B and SMAD3 (compare Fig. 1A).

Interestingly, the highly invasive Panc1, PaCa3, MiaPaCa2 and MDA-MB-231 cells all exhibit high autocrine TGF-b production (Ref. #10 and #23 in the revised version, PaCa3: our own unpublished observation), while the poorly invasive lines do not (i.e. Colo357: Ref. 23). In Panc1 and other TGF-β-sensitive PDAC lines, TGF-β has been shown to upregulate its own expression and to act in an autocrine manner to maintain tumor cell invasion, while in cell lines with defects in TGFBR2 (MiaPaCa2) or loss of SMAD4 expression (BxPC3, Capan1, Capan2) TGF-β failed to autoinduce its own expression (Ref. #10). As outlined under point #1, we therefore assume that in order for SMAD3 to suppress migration, cells need to harbor a functional TGF-b pathway and/or exhibit autocrine TGF-b1 production. We have added a few sentences on this issue to the Discussion and replaced the term “TGF-β-dependent/independent” by “activation-dependent/independent” in Fig. 7. We also changed the title to include “in highly metastatic pancreatic cancer cells”. For a possible mechanistic explanation, see below.

Is there any insight how RAC1B induces the expression of SMAD3?

Response: Yes, we have preliminary data to indicate that RAC1B promotes synthesis and secretion of TGF-b1 and that SMAD3 is induced by RAC1B through intermittent production of autocrine TGF-b1. This is inferred from the observations that knockdown of RAC1B in Panc1 cells decreased the levels of secreted TGF-b1 in culture supernatants, while knockdown of TGF-b1 decreased SMAD3 mRNA and protein levels. Moreover, we observed that RAC1B itself is downregulated by TGF-b1 which suggests that both form a negative feedback loop. These data although being preliminary at this point could help to explain why TGF-b-insensitive cells have higher protein levels of RAC1B and SMAD3 and why the antimigratory effect of SMAD3 is seen only in cell lines with functional TGF-b signaling and (high) autocrine TGF-b production. These considerations have been added to the Discussion section.

Could the authors compare which siRNA-mediated inhibition: RAC1B or SMAD3 or both together is more effective to reverse the anti-migratory effects?

Response: To clarify this issue we have carried out additional migration assays with Panc1 and PaCa3 cells transfected in parallel with only RAC1B siRNA, only SMAD3 siRNA, or a combination of both. For Panc1 cells, we have quantified the migration data and included them as inset in Figure 3B. The data show that in Panc1 cells RAC1B and SMAD3, or the combination of both, are equally potent in derepressing the migratory activity.

In line 196, authors have written that the performed the experiment in two cell lines, however, only experiments performed in Panc-1 are shown.

Response: This is true and we apologize for this mistake. This second cell line was meant to be the breast cancer cell line MDA-MB-231. However, in response to a request from Reviewer 2 we have removed the data with these cells from the revised version and, accordingly, the mention of these cells in line 196.

The authors should reconsider to change this title “Ectopic Expression of SMAD3 Rescues Panc1 Cells from the RAC1B Knockdown-Induced Increase in Migratory Activity whereas Knockdown of SMAD3 Rescues Panc1 Cells from the RAC1B Overexpression-Induced Decrease in Migratory Activity” is long and difficult to understand.

Response: We have rephrased this subheading (“SMAD3 Mediates the Antimigratory Effect of RAC1B”) to enhance clarity.

How the authors can explain the differences found between the SIS3 drug and the pSMAD3-mut3A (Figures 5A and 5C)? If I well understood, in both cases there is an inhibition of the phosphorylation of the C-terminal of the protein. However, both approaches have the same effect after TGF-b treatment. This explanation is not clear “these data suggest that in order to antagonize basal migration, C-terminal phosphorylation of SMAD3 is dispensable. However, SMAD3-C phosphorylation is nevertheless required to promote cell migration in response to TGF-b1 stimulation.”

Response: SIS3 only inhibits C-terminal phosphorylation of SMAD3 in response to TGF-b1 treatment. Therefore, in the absence of TGF-b (under basal conditions) it has no effect (no statistically significant difference between the red and green curves in panel 5A). In contrast, the SMAD3-mut3a mutant combines two functional activities, the ability to inhibit C-terminal phosphorylation of SMAD3 in a dominant-negative fashion in response to TGF-b1 (like SIS3) and the ability to relieve the increase in migration upon RAC1B knockdown (same effect as wild-type SMAD3, compare with Figure 4A). We have rephrased the explanation accordingly.

In vivo assays in immunocompetent mice would be very informative to determine the role of the axis RAC1B/SMAD3/BGN in cell migration and metastasis. Besides the anti-migratory effect that the author shown in the manuscript, previously it has been shown that Smad3 has different roles in the tumor development, inducing immunosuppression or EMT. An in vivo model could clarify if promoting RAC1B and inducing SMAD3 and BGN expression may be a therapeutic strategy for PDAC.

Response: This is certainly a good suggestion and these experiments are currently being scheduled. So far, we could not perform them due to high costs and lack of appropriate animal facilities. As a word of caution, it will be difficult to discriminate between the effects of non-phosphorylated SMAD3 (anti-migratory) and C-terminally phosphorylated SMAD3 (pro-migratory) in vivo. The outcome will seriously depend on the tissue concentrations of TGF-b (and those of related factors able to activate SMAD3). Given the generally high TGF-b concentration in PDAC tissues, the pro-invasive effects of SMAD3 may predominate. See also point #1 of Reviewer 2 that is related to this issue.

Reviewer 2 Report

Merits: Detailed investigation of Rac1b/SMAD3 antimigratory regulatory axis with sophisticated controls and nuances. Demonstration that other SMAD family members (e.g. SMAD2) are not involved in this activity. Additionally, investigations with genetic modification (of SMAD3 C-terminal domain) and usage pharmacological inhibitors were aptly used to characterize the pathway. Interesting observation about inverse relationship between basal Rac1b and SMAD3 levels and "metastatic potential" and tumor grade across several pancreatic cancer cell lines.

Shortcomings/suggestions: It would be interesting to see if the observation of Rac1b/SMAD3 levels holds true on a large scale and relevant to patent pathophysiology, i.e. microarray study. However, for this manuscript, it is reasonable to suggest using bioinformatics resources (e.g. TCGA) with preexisting data to see if low Rac1b/SMAD3 levels are correlated with poorer PFS, OS, higher tumor grade, etc; this would not be time-consuming or technically challenging, and could considerably strengthen the study. Redo all statistical analyses. There are often multiple t-tests performed on the same data; in the case of Fig 1A, the chance of committing a type I error is over 43%, rather than the intended 5%, due to error propagation of performing nearly a dozen tests. An ANOVA test is far preferable to such repetitive t-tests, though for some of the less offending experiments, you could do an ANOVA incorporating both Bartlett and Brown-Forsythe tests to verify both Gaussian distribution of residuals and equal variances among SDs, followed by several (not a dozen) t-tests with Holm-Sidak correction for multiple comparisons. It is never explicitly stated which statistical evaluation is used for real-time migration assays, though the only one mentioned is the t-test; this suffers from a similar issue, to an even greater extent. Some of the migration assays could utilize a two-way ANOVA, some only a one-way ANOVA could be employed. Good job using SD instead of SEM for your error bars, however, there are some that lack error bars entirely, mostly on controls (e.g. Fig 1B, Fig 2A, Fig 6A,C) which also have statistical tests performed on them, which is a faux pas, especially given specific the test performed. Western blots should really be regarded as semi-quantitative, at best; however, several of the blots (e.g. Fig 1B, Fig 2C, Fig 3A, perhaps others) have clearly oversaturated bands, which can be used for qualitative analysis, but cannot be used for "quantification". That being said, it would make some of the data appear to be even better, since the oversaturated bands means the quantification is actually underestimating them. Fig 4C and possibly Fig 5D the images of the blots have been over-manipulated for gamma/contrast; some bands are cropped too closely, such as Fig 4C. Some of the insets are too small to decipher, particularly Fig 6B,D. In most regards, this is a single cell line study, also a single phenotype. It would be more intriguing if invasiveness were also investigated, but that is a fairly major undertaking. The study could be considerably strengthened by the addition of confirmatory cell migration experiments with other pancreatic cancer cell lines. On that note, it is unclear why they included a breast cancer cell line (MDA-MB-231), seemingly an afterthought. Furthermore, the paltry amount of data relating MDA-MB-231 is not particularly encouraging (Fig S2A), and is likely not statistically significant with the correct analysis performed. I'd remove the few mentions of MDA and breast cancer, as they add nothing.
A third of all citations used are self-citations: 11/34 have at least one of the current manuscript's authors. This isn't such a niche subject that warrants such citation bias.

Author Response

Merits: Detailed investigation of Rac1b/SMAD3 antimigratory regulatory axis with sophisticated controls and nuances. Demonstration that other SMAD family members (e.g. SMAD2) are not involved in this activity. Additionally, investigations with genetic modification (of SMAD3 C-terminal domain) and usage pharmacological inhibitors were aptly used to characterize the pathway. Interesting observation about inverse relationship between basal Rac1b and SMAD3 levels and "metastatic potential" and tumor grade across several pancreatic cancer cell lines.

Shortcomings/suggestions:

It would be interesting to see if the observation of Rac1b/SMAD3 levels holds true on a large scale and relevant to patent pathophysiology, i.e. microarray study. However, for this manuscript, it is reasonable to suggest using bioinformatics resources (e.g. TCGA) with preexisting data to see if low Rac1b/SMAD3 levels are correlated with poorer PFS, OS, higher tumor grade, etc; this would not be time-consuming or technically challenging, and could considerably strengthen the study.

Response: We thank the reviewer for this valuable suggestion. Regarding RAC1B we have observed previously that high expression in PDAC tissues correlated with prolonged patient survival (see Ref. 6, mentioned in the Introduction, lines 61-62), which is in agreement with the functional antagonism between RAC1 and RAC1B in vitro. In addition and as requested, we have used the GEPIA database (http://gepia.cancer-pku.cn/) to test for associations. Both the related RAC1 and SMAD3 are highly overexpressed in pancreatic carcinoma (gene expression profile in PAAD (=PDAC) tumor vs. normal tissue is 288.3 to 42.95 for RAC1 and 25.01 to 6.94 for SMAD3 (no data are available for RAC1B). Hence, for both genes PAAD is among the three tumor types with the highest overexpression. Moreover, high expression of both RAC1 and SMAD3 is correlated with poorer OS and PFS. The worse prognosis of the “high-SMAD3” group, however, is likely due to formation of phospho-SMAD3-C by TGF-b known to mediate its tumor-promoting effects. Unfortunately, there are no bioinformatic resources that separately assemble clinical data for non-phosphorylated SMAD3 and phosphorylated SMAD3. Hence, an association with survival or tumor grade with only the non-phosphorylated of SMAD3 is not possible. As mentioned under point #10 of Reviewer 1, we assume that the generally high TGF-b concentration in PDAC tissues will favor phosphorylation and hence the pro-invasive effects of SMAD3.

Redo all statistical analyses. There are often multiple t-tests performed on the same data; in the case of Fig 1A, the chance of committing a type I error is over 43%, rather than the intended 5%, due to error propagation of performing nearly a dozen tests. An ANOVA test is far preferable to such repetitive t-tests, though for some of the less offending experiments, you could do an ANOVA incorporating both Bartlett and Brown-Forsythe tests to verify both Gaussian distribution of residuals and equal variances among SDs, followed by several (not a dozen) t-tests with Holm-Sidak correction for multiple comparisons. It is never explicitly stated which statistical evaluation is used for real-time migration assays, though the only one mentioned is the t-test; this suffers from a similar issue, to an even greater extent. Some of the migration assays could utilize a two-way ANOVA, some only a one-way ANOVA could be employed. Good job using SD instead of SEM for your error bars, however, there are some that lack error bars entirely, mostly on controls (e.g. Fig 1B, Fig 2A, Fig 6A,C) which also have statistical tests performed on them, which is a faux pas, especially given specific the test performed.

Response: We have discussed this issue with our biostatisticians. The qPCR and immunoblot data show the means ± SD of at least three independent and identically performed experiments. Statistical analyses was done based on data matching from different experiments using the Wilcoxon-test for dependent random sampling and therefore faithfully assesses interassay variability. The figures with migration assays show the results from the most representative experiment out of a series of at least three independent experiments and show the means ± SD of 3-4 parallel wells. Statistical analyses was done for the cells of the control group compared to the cells of the test group for each time point. Here we have employed the Mann-Whitney-U-test for independent random samples as done previously (Ungefroren et al. 2017, Mol Pharmacol 92:519).

Errors bars in the controls (i.e. in Figures 1B, 2A, 6A,C) have not been indicated because data represent the mean of at least 3 experiments following normalization to have the controls of each experiment set at 1.0 prior to calculations of SDs of the test group. Hence, the SD of the control group is always zero. This important piece of information was added to the respective figure legends.

Western blots should really be regarded as semi-quantitative, at best; however, several of the blots (e.g. Fig 1B, Fig 2C, Fig 3A, perhaps others) have clearly oversaturated bands, which can be used for qualitative analysis, but cannot be used for "quantification". That being said, it would make some of the data appear to be even better, since the oversaturated bands means the quantification is actually underestimating them.

Response: It should be noted that the bands used for quantification are not those depicted in the figures. The ChemiDoc XRS+ System combined with the Image Lab software from BioRad that we were using for chemoluminescent detection of proteins detects a wide range of sample concentrations without saturating the most concentrated band, enabling linear quantitation. For more information, please visit http://www.bio-rad.com/webroot/web/pdf/lsr/literature/Bulletin_5837.pdf.

Fig 4C and possibly Fig 5D the images of the blots have been over-manipulated for gamma/contrast; some bands are cropped too closely, such as Fig 4C. Some of the insets are too small to decipher, particularly Fig 6B,D.

Response: We should stress that we did not manipulate any of the blot images. The blot in the inset in Fig. 4C contains an irrelevant lane with a strong signal left to the “pR1B” lane. For this reason, we had to crop it so closely. As requested, we have enlarged the insets in Figures 3-6. In response to a request from the Associate Editor, we had provided on November 6, 2019, a Supplementary data file containing the uncropped immunoblots along with molecular weight markers and densitometric readings of band intensities. I suppose that this file will be published along with this manuscript.

In most regards, this is a single cell line study, also a single phenotype. It would be more intriguing if invasiveness were also investigated, but that is a fairly major undertaking. The study could be considerably strengthened by the addition of confirmatory cell migration experiments with other pancreatic cancer cell lines.

Response: This issue was also raised by Reviewer 1. Please, see my detailed responses to requests #1 and #4 of Reviewer 1.

On that note, it is unclear why they included a breast cancer cell line (MDA-MB-231), seemingly an afterthought. Furthermore, the paltry amount of data relating MDA-MB-231 is not particularly encouraging (Fig S2A), and is likely not statistically significant with the correct analysis performed. I'd remove the few mentions of MDA and breast cancer, as they add nothing.

Response: MDA-MB-231 cells have been used because this cell line is highly metastatic and resemble Panc1 and PaCa3 cells with respect to TGF-b sensitivity and autocrine TGF-b production (see point #4 of Reviewer 1 for why this is important). As suggested, we have removed the data on MDA-MB-231 cells and have replaced them with data from another PDAC-derived cell line, PaCa3 (please also see request #1 of Reviewer 1). The intention of including data from a breast cancer cell line was to provide evidence that the regulatory cascade involving RAC1B and SMAD3 is not confined to PDAC but also operates in other tissues/tumor types.

A third of all citations used are self-citations: 11/34 have at least one of the current manuscript's authors. This isn't such a niche subject that warrants such citation bias.

Response: We are well aware of this “problem”. However, this is almost inevitable since our group is the only one studying RAC1B in TGF-b signaling and cell motility and referring to our previous work is important for the reader to obtain sufficient background information. However, we have removed two self-citations (#8 and #11 in the original version) and have referenced 8 other studies (highlighted in yellow in the revised version) in order to weaken this bias. The new references have been included in response to a request from Reviewer 1 (see point #4).

Round 2

Reviewer 1 Report

I thank the author for their reply to my comments.